# A Review on Multi-GNSS for Earth Observation and Emerging Applications

**Shuanggen Jin [1,2]**, **Qisheng Wang [3]** and **Gino Dardanelli [4,***

1  School of Surveying and Land Information Engineering, Henan Polytechnic University, Jiaozuo 454000, China
2  Shanghai Astronomical Observatory, Chinese Academy of Sciences, Shanghai 200030, China
3  College of Civil Engineering, Xiangtan University, Xiangtan 411105, China
4  Department of Engineering, University of Palermo, Viale delle Scienze, 90128 Palermo, Italy
*  Correspondence: gino.dardanelli@unipa.it; Tel.: +39-019-2389-6228

**Abstract:** Global Navigation Satellite System (GNSS) has drawn the attention of scientists and users all over the world for its wide-ranging Earth observations and applications. Since the end of May 2022, more than 130 satellites are available for fully global operational satellite navigation systems, such as BeiDou Navigation Satellite System (BDS), Galileo, GLONASS and GPS, which have been widely used in positioning, navigation, and timing (PNT), e.g., precise orbit determination and location-based services. Recently, the refracted, reflected, and scattered signals from GNSS can remotely sense the Earth's surface and atmosphere with potential applications in environmental remote sensing. In this paper, a review of multi-GNSS for Earth Observation and emerging application progress is presented, including GNSS positioning and orbiting, GNSS meteorology, GNSS ionosphere and space weather, GNSS-Reflectometry and GNSS earthquake monitoring, as well as GNSS integrated techniques for land and structural health monitoring. One of the most significant findings from this review is that, nowadays, GNSS is one of the best techniques in the field of Earth observation, not only for traditional positioning applications, but also for integrated remote sensing applications. With continuous improvements and developments in terms of performance, availability, modernization, and hybridizing, multi-GNSS will become a milestone for Earth observations and future applications.

**Keywords:** GNSS; GNSS meteorology; GNSS ionosphere; GNSS-Reflectometry; GeoHazards

## 1. Introduction

The Global Navigation Satellite System (GNSS) has been developed rapidly and attracted increasing global attentions for its wide-ranging Earth monitoring and investigatory applications. Since the end of May 2022, more than 130 satellites of fully global operational satellite navigation systems are available, such as China's BeiDou Navigation Satellite System (BDS) [1], the European Union's Galileo [2], Russia's GLObal NAvigation Satellite System (GLONASS) [3] and the United States' Global Positioning System (GPS) [4]. Furthermore, many other regional GNSS systems are available, e.g., the Indian Regional Navigation Satellite System (IRNSS/NavIC), the Japanese Quasi-Zenith Satellite System (QZSS), and the Regional South Korean Positioning System (KPS), which hold massive potential applications in the scientific community. In terms of applications, the ground-based and space-borne GNSS receivers can measure the ionospheric total electron content (TEC) for the global ionospheric climate and space weather (GNSS-Ionosphere). The dense TEC observations can record ionospheric perturbations, due to earthquakes, tsunamis, volcanos, typhoons, or geomagnetic storms, and eclipses [5–7]. The GNSS Reflectometry (GNSS-R) from a low Earth-orbiting satellite can retrieve environmental parameters over the sea ice, lands, and oceans [8,9].

Nowadays, multi-GNSS represents one of the best techniques in the field of Earth observations. Of course, with continuous improvements and developments in terms of

performance, availability, modernization, and hybridizing, GNSSs will be involved in more future applications. The aim of this review is to present the latest state of multi-GNSS for Earth observations and emerging applications, including the following, not fully exhaustive, troposphere and ionosphere observations; modeling and assimilation from ground-based and space-borne GNSS observations; theory and methods of multi-GNSS near real-time kinematic (RTK) positioning; precise point positioning (PPP) and PPP-RTK; GNSS-Reflectometry applications; geohazard observation and warning from GNSS; errors, systematic effects and noise in GNSS solutions; and surface-loading GNSS observations from atmosphere, hydrology and loading.

In this paper, a review of multi-GNSS for Earth Observation and emerging application progress is presented, including GNSS positioning and orbiting, GNSS meteorology, GNSS ionosphere and space weather, and GNSS-Reflectometry as well as GNSS earthquake monitoring and GNSS integrated techniques for land and structural health monitoring. In Section 2, BDS/GNSS theory, methods and error resources are presented, as well as multi-GNSS observations. GNSS emerging applications are reviewed in details in Section 3. A summary and prospective are given in Section 4.

## 2. BDS/GNSS Techniques and Observations

### 2.1. BDS/GNSS Techniques

As it is widely known, GPS and GLONASS were the first two global positioning systems used in different fields. Their Earth observation comes mainly from dual frequency signals. Global Positioning System (GPS) was originally Navstar GPS, which is a satellite-based radionavigation system by USA. GPS provides geolocation and time information to a GPS receiver anywhere on or near the Earth with four or more GPS satellites. GPS can provide precise positioning capabilities to military, civil, and commercial users around the world. GPS uses code division multiple access (CDMA), while GLONASS uses frequency division multiple access (FDMA). With the development of science, technology and society, GPS and GLONASS were implemented with a new generation of satellites, and three frequency signals were established. In particular, CDMA technology was used by GLONASS for its new signals [10]. Moreover, Galileo, as the main provider of civil services, has quickly developed in recent years to provide global services with multi-frequency signals [11]. After three-step strategic developments, China's BeiDou Global Navigation Satellite System (BDS-3) has been able to provide global services with more signals [12,13]. All constellations of GPS, GLONASS and Galileo adopt the constellation of medium Earth orbit (MEO), while the constellation of BDS is composed of MEO satellites, geostationary orbit (GEO) satellites and inclined geosynchronous orbit (IGSO) satellites [1–4]. Based on existing global services, BDS has improved service performances within the region and the global. The number of GNSS satellites are shown in Table 1 with satellites types and signals. Nowadays, multi-frequency multi-GNSS technology is used for Earth observation. Table 2 shows the GNSS multi-frequency signals from the Receiver Independent Exchange (RINEX) format [14]. Note that this study only presents and discusses the publicly available signals.

**Table 1.** Number of multi-GNSS satellites in orbit (until 2022) [1–4].

| System | Block | Signal | Number of Operational Satellites |
|---|---|---|---|
| GPS | IIR | L1 L2 | 7 |
| | IIR-M | L1 L2 | 7 |
| | IIF | L1 L2 L5 | 12 |
| | III/IIIF | L1 L2 | 5 |
| GLONASS | M | G1 G2 | 22 |
| | K | G1 G2 G3 | 1 |
| Galileo | IOV | E1 E6 E5a/b/ab | 3 |
| | FOC | E1 E6 E5a/b/ab | 19 |

**Table 1.** *Cont.*

| System | Block | Signal | Number of Operational Satellites |
|---|---|---|---|
| BDS-2 | MEO | B1 B2 B3 | 3 |
|  | IGSO | B1 B2 B3 | 7 |
|  | GEO | B1 B2 B3 | 5 |
| BDS-3 | MEO | B1 B3 B1C B2 a/b | 24 |
|  | IGSO | B1 B3 B1C B2 a/b | 3 |
|  | GEO | B1 B3 | 2 |

**Table 2.** Multi-frequency multi-GNSS signals [14].

| System | Freq. Band | Frequency/MHz | Observation Codes |
|---|---|---|---|
| GPS | L1 | 1575.42 | C1C C1S C1L C1X C1P C1W C1Y C1M |
|  | L2 | 1227.60 | C2C C2D C2S C2L C2X C2P C2W C2Y C2M |
|  | L5 | 1176.45 | C5I C5Q C5X |
| GLONASS | G1 | 1602 + k × 9/16 k = −7 . . . + 12 | C1C C1P |
|  | G2 | 1246 + k × 7/16 | C2C C2P |
|  | G3 | 1202.025 | C3I C3Q C3X |
| Galileo | E1 | 1575.42 | C1A C1B C1C C1X C1Z |
|  | E5a | 1176.45 | C5I C5Q C5X |
|  | E5b | 1207.140 | C7I C7Q C7X |
|  | E5 (E5a + E5b) | 1191.795 | C8I C8Q C8X |
|  | E6 | 1278.75 | C6A C6B C6C C6X C6Z |
| BDS-2 | B2 | 1207.140 | C7I C7Q C7X |
| BDS-2/3 | B1 | 1561.098 | C2I C2Q C2X |
|  | B3 | 1268.52 | C6I C6Q C6X |
| BDS-3 | B1C | 1575.42 | C1D C1P C1X |
|  | B1A | 1575.42 | C1S C1L C1Z |
|  | B2a | 1176.45 | C5D C5P C5X |
|  | B2b | 1207.140 | C7D C7P C7Z |
|  | B2 (B2a + B2b) | 1191.795 | C8D C8P C8X |
|  | B3A | 1268.52 | C6D C6P C6Z |

*2.2. GNSS Observation Equations*

The GNSS pseudo-range and carrier phase observations can be expressed as [15]

$$\begin{cases} p_{r,j}^s = \rho_r^s + c(dt_r - dt^s) + T_r^s + I_{r,j}^s + c(d_{r,j} - d_j^s) + \varepsilon_p \\ \varphi_{r,j}^s = \rho_r^s + c(dt_r - dt^s) + T_r^s - I_{r,j}^s + \lambda_j w_r^s + \lambda_j N_{r,j}^s + \lambda_j(b_{r,j} - b_j^s) + \varepsilon_\varphi \end{cases} \tag{1}$$

where the superscript s denotes a GNSS satellite; the subscript $r$ and $j$ denote the receiver and the frequency; $P_{r,j}^s$ denotes the observed pseudo-range on $j$th frequency in meters; $\varphi_{r,j}^s$ is the corresponding carrier phase; $\rho_r^s$ denotes the geometrical range from phase centers of the satellite to receiver antennas at the signal transmitting and receive time in meters; $c$ denotes the vacuum speed of light in meters per second; $d$tr is the receiver clock offset in seconds; $d$ts is the satellite clock offset in seconds; $T_r^s$ is the slant tropospheric delay in meters; $I_{r,j}^s$ is the ionospheric delay on $j$th frequency in meters; $d_{r,j}$ and $d_j^s$ are the code biases of the receiver and the satellite in seconds; $\lambda_j$ is the wavelength of carrier phase on the $j$th frequency in meters; $w_r^s$ is the phase wind-up delay in cycles; $N_{r,j}^s$ is the integer ambiguity on the $j$th frequency in cycles; $b_{r,j}$ and $b_j^s$ are the uncalibrated phase delays (UPDs) for receiver and satellites in cycles, respectively; $\varepsilon_p$ and $\varepsilon_\varphi$ are the pseudo range and carrier phase observation noises including multipath in meters, respectively.

In order to eliminate some parameters, the original observations of GNSS can be linearly combined. There are several common combinations, such as geometry-free (GF) [16], ionosphere-free (IF) [17], wide lane (WL) and narrow lane (NL) [18] combinations. GF combination eliminates the frequency independent parameters and it is used for ionospheric modeling. The IF combination is often used for precise single point positioning, while WL and NL combinations can be used for cycle-slip detection [16–18].

### 2.3. GNSS Positioning Methods

There are two common methods for GNSS positioning, namely differential positioning, and precise point positioning (PPP). Differential positioning is a relative positioning technology, which needs to set up a reference station with a known position and observes synchronously with the user [19]. PPP is an absolute positioning technique with removing or estimating GNSS errors, which provides a high level of position accuracy by a single receiver observation [20].

For GNSS differential positioning, there are four different types of position differentials: pseudo range differential, carrier phase smoothing, pseudo range differential and carrier phase differential. Real Time Kinematic (RTK), used to obtain precise positioning results, adopts carrier phase differential technology. However, the base stations need to be set up for RTK, so the operation mode is not flexible and the cost is relatively high. Moreover, with increase of the distance between users and reference stations, its positioning accuracy becomes significantly reduced. Comparing the two methods, since non-difference observation is used in PPP, the latter provides more advantages in terms of operation mode and cost [21,22].

The GNSS PPP method is based on three common models, the ionosphere-free combination mode, the Uofc model and the uncombined model [23]. Among these, the most used ionospheric-free model is able to eliminate the ionospheric parameters through the combination of dual frequency pseudo range and carrier phase observations [17]:

$$\begin{cases} P_{IF} = \alpha \cdot p^s_{r,i} + \beta \cdot p^s_{r,j} \\ \Phi_{IF} = \alpha \cdot \varphi^s_{r,i} + \beta \cdot \varphi^s_{r,j} \end{cases} \tag{2}$$

where $\alpha = f_i^2 / (f_i^2 - f_j^2)$, $\beta = -f_j^2 / (f_i^2 - f_j^2)$.

The Uofc model, based on the ionospheric-free model with two frequency phase observations, involves the ionospheric free combination composed of phase and pseudo range observations at each frequency [22–25]:

$$\begin{cases} P_{UofC,i} = (p^s_{r,i} + \varphi^s_{r,i})/2 \\ P_{UofC,j} = (p^s_{r,j} + \varphi^s_{r,j})/2 \\ \Phi_{UofC,ij} = \Phi_{IF} \end{cases} \tag{3}$$

The above two models both eliminate the ionospheric parameters through the combination of observation equations, but also enlarge the observation errors. In the last few years, the uncombined model, by using the original observation equation shown in Equation (1), has received more attentions, and thus has been involved in several applications, such as positioning, timing, and tropospheric retrieval applications [26,27]. The three PPP models are theoretically equivalent [22,25]. With the development of multi-frequency and multi-GNSS, the ionospheric-free model and Uofc model have been extended to handle multi-frequency signals. More PPP models can be extended based on the above three models, considering ionospheric parameter corrections or weight [11,28–30].

### 2.4. Main Error Sources

The main errors in GNSS observations [31,32] are, for instance, those related to the GNSS satellite, to the GNSS receiver, to signal propagation among others, as shown in Figure 1, including satellite orbit error, satellite clock offset, phase center offset, receiver

clock offset, receiver hardware delay, antenna phase center offset, ionospheric and tropospheric delay, relativistic effect and multipath.

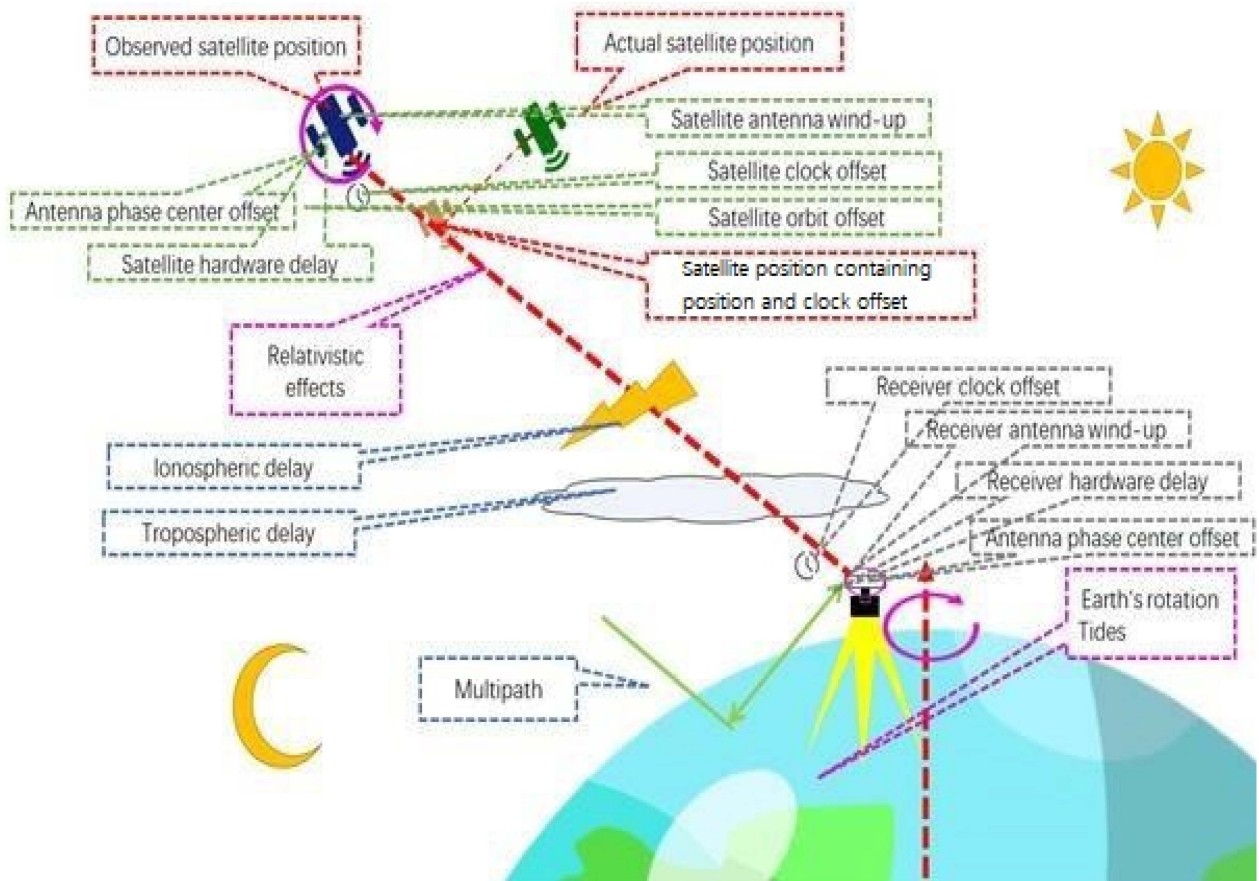

**Figure 1.** GNSS error source.

### 2.5. Multi-GNSS Observations

As a volunteer association, the International GNSS Service (IGS) has provided the highest-quality GNSS data to users for the last twenty years. With the development of newly established global and regional navigation satellite systems, IGS conducted the multi-GNSS experiment (MGEX) project to collect and analyze observations of the new systems and signals [33,34]. The MGEX network started in 2012, and grew rapidly in the following years. In May 2022, the number of IGS and MGEX stations rose to more than 500 (Figure 2). All stations can track GPS, while about 450 stations are available for GLONASS. There are approximately 370 and 310 stations to track Galileo and BDS, respectively. Table 3 summarizes the information of GNSS receivers in MGEX (until May 2022). As shown, most of the available receivers can track multiple GNSS. These stations provide sufficient guarantee for GNSS Earth observation data. Note that there are other public agencies and some private vendors, which can provide single- or multi-GNSS observations, such as the GPS Earth Observation Network System (GEONET), operated by the Geospatial Information Authority of Japan (GSI).

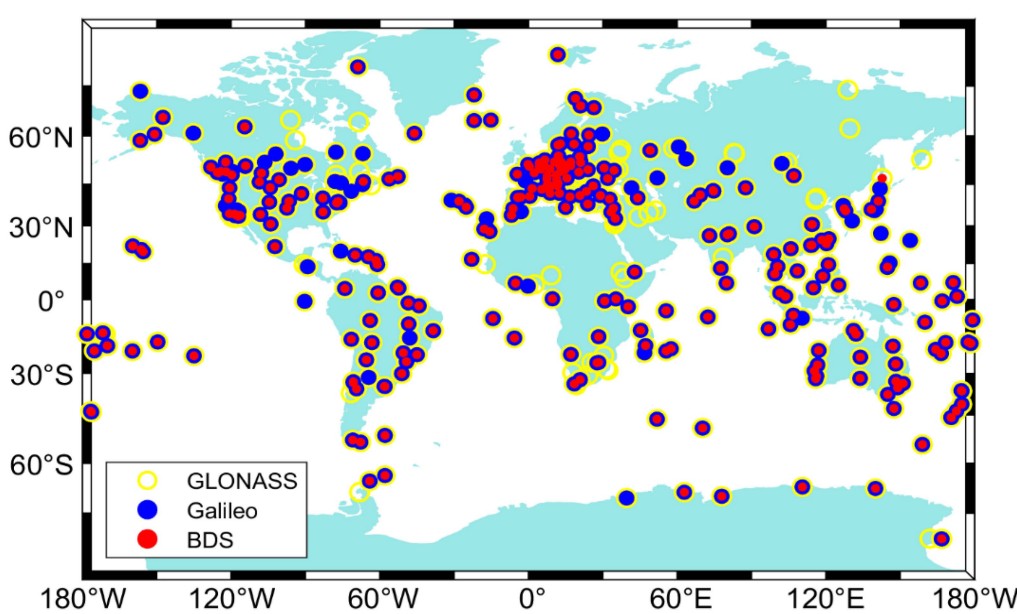

**Figure 2.** IGS/MGEX stations in May 2022.

**Table 3.** Information of GNSS receiver in MGEX (until May 2022).

| ID | Receiver | Type | Trackable Satellite | Stations |
|----|----------|------|---------------------|----------|
| 1 | ASHTECH | UZ-12, Z-XII3, Z-XII3T | GPS | 16 |
| 4 | CHC | P5E | GPS + GLO + GAL + BDS | 1 |
| 5 | JAVAD | TR_G3TH, TRE_3, TRE_3 DELTA, TRE_3L DELTA, TRE_3N DELTA, TRE_G2T DELTA, TRE_G3T DELTA, TRE_G3TH DELTA, | GPS + GLO + GAL + BDS | 74 |
| 6 | JPS | EGGDT, LEGACY | GPS + GLO | 7 |
| 7 | LEICA | GR10, GR25, GR30, GR50, GRX1200 | GPS + GLO + GAL + BDS | 57 |
| 8 | NOV | OEM4-G2, OEM6, OEMV3 | GPS | 23 |
| 9 | SEPT | ASTERX4, POLARX2, POLARX3ETR, POLARX4TR, POLARX5, POLARX5E, POLARX5S, POLARX5TR, | GPS + GLO + GAL + BDS | 153 |
| 10 | STONEX | SC2200 | GPS + GLO + GAL + BDS | 1 |
| 11 | TPS | LEGACY, NETG3, NET-G3A, NET-G5 | GPS + GLO | 20 |
| 12 | TRIMBLE | 5700, ALLOY, NETR5, NETR8, NETR9, NETRS, R9S | GPS + GLO + GAL + BDS | 160 |
| | | | Total | 512 |

## 3. GNSS Emerging Applications

### 3.1. GNSS Positioning and Orbiting

Single-frequency GNSS Precise Point Positioning (PPP) can achieve a centimeter-decimeter accuracy level and multi-frequency GNSS PPP can obtain a millimeter-centimeter level when the carrier phase ambiguities converge. Figure 3 shows the errors of single-, dual-, triple- and quad-frequency static BDS PPP at the iGMAS station KUN1 in the north, east and vertical directions on DOY 16, 2019 [28]. The positioning performance

was compared with the iGMAS products. From the comparison of single-, dual-, triple- and quad-frequency BDS PPP performances, the multi-frequency BDS signals were able to greatly improve the positioning performance, particularly for quad-frequency BDS observations. It also showed that the positioning errors were decreased with increasing observation time.

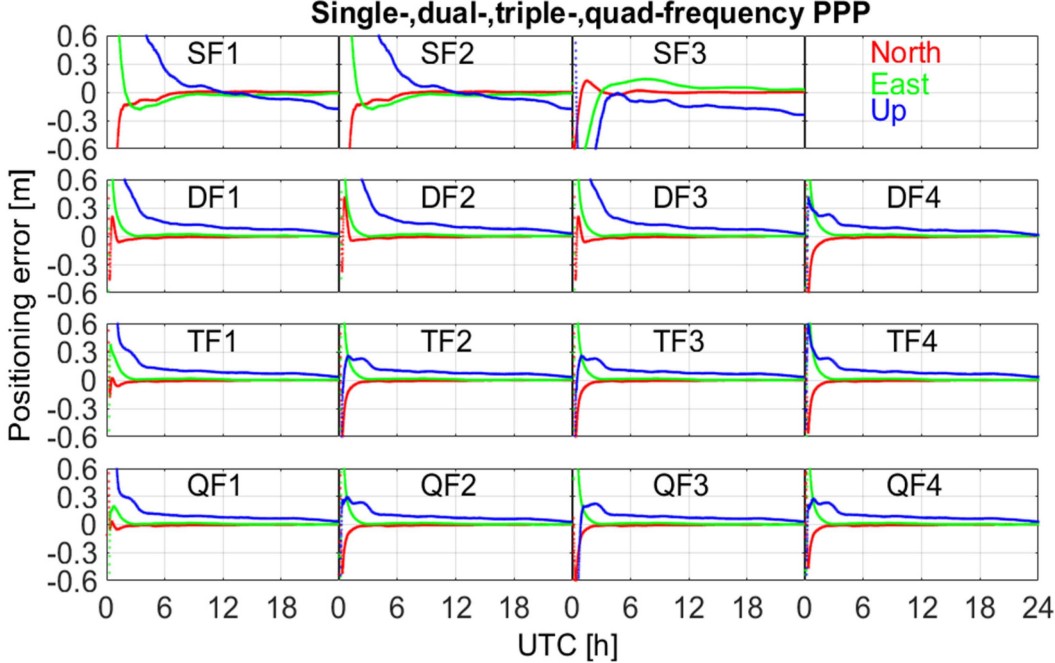

**Figure 3.** Errors of single-, dual-, triple- and quad-frequency BDS PPP at the iGMAS station KUN1 in the north, east and vertical direction on DOY 16, 2019 [28].

Precise positioning and precise orbit determination (POD) can be estimated from GNSS observations. For example, the potential periodicity of empirical acceleration in the Haiyang 2B (HY-2B) POD was identified by spectral analysis. Using over one year of satellite laser ranging (SLR) measurements, a 5.2% improvement in the orbit solution of the refined model was demonstrated and validated. After application of the in-flight calibration of the GPS antenna, a 26% reduction in the root mean square (RMS) of SLR residuals was achieved for the reduced-dynamic (RD) orbit solutions, and the carrier phase residuals were clearly reduced. The integer ambiguity resolution of HY-2B led to strong geometric constraints for the estimated parameters, and a 15% improvement in the SLR residuals could be inferred when compared with the float solution [35].

One year's data collected by the Gravity Recovery and Climate Experiment Follow-On (GRACE-FO) mission and GPS precise products provided by the Center for Orbit Determination in Europe (CODE) were analyzed. The precise orbit, generated by the Jet Propulsion Laboratory (JPL), independent SLR, and K-band ranging (KBR), measurements were utilized to assess the orbit accuracy. More than 98% of single difference (SD) wide-lane (WL) and 95% of SD narrow-lane (NL) ambiguities were fixed, which confirmed the good quality of the bias products and the correctness of the SD ambiguity resolution (AR) method [36,37].

### 3.2. GNSS Meteorology

Tropospheric delay is one of the most common GNSS positioning errors. Nowadays, the zenith tropospheric delay (ZTD) can be estimated from GNSS observations, which can be transferred into precipitable water vapor (PWV) for meteorological applications. Recently, PWV was estimated and analyzed at 377 GNSS stations from the infrastructure construction of national geodetic datum modernization and Crustal Movement Observation

Network of China (CMONC), which is one of the most important Continuously Operating Reference Station (CORS) networks in the world. Further PWV was obtained from GPS observations and meteorological data from 2011 to 2019. The PWV had improved accuracy when compared with the Bevis model. Bevis et al. [38] pioneered the concept of GPS meteorology and obtained global surface temperature and weighted average temperature from more than 8000 radiosonde stations in America. They also described the linear relationship coefficient and the specific process of GPS water vapour inversion on the ground. Furthermore, the daily and monthly average values, long-term trend, and change characteristics of the PWV were analyzed, using the high-precision inversion model. The results showed that the averaged PWV was higher in Central–Eastern China and Southern China and lower in Northwest China, Northeast China, and North China (Figure 4). The PWV was increasing in most parts of China, while some PWVs in North China showed a downward trend [39].

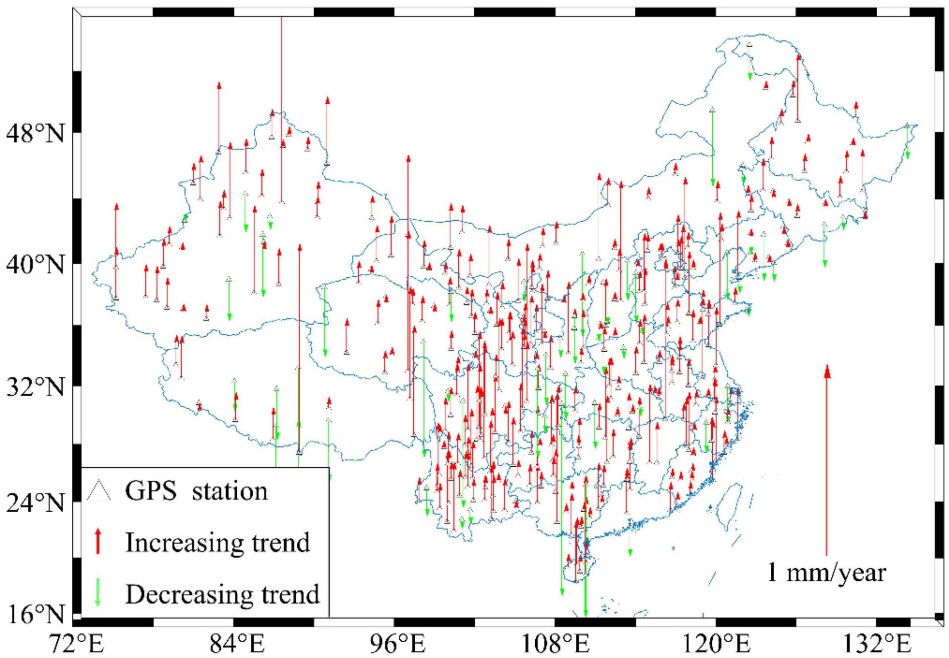

**Figure 4.** Long-term variation trend of the GNSS PWV from 2011 to 2018 [39].

In addition, pressure, water vapor pressure, temperature, and weighted mean temperature (Tm) play an important role in GNSS meteorological applications. A new approach was introduced to develop an empirical tropospheric delay model, named the China Tropospheric (CTrop) model, to provide meteorological parameters based on the sliding window algorithm, using radiosonde data as reference values to validate the performance of the CTrop model, which was compared to the canonical Global Pressure and Temperature 3 (GPT3) model. The accuracy of the CTrop model in regards to pressure, water vapor pressure, temperature, and weighted mean temperature were 5.51 hPa, 2.60 hPa, 3.09 K, and 3.35 K, respectively, achieving an improvement of 6%, 9%, 10%, and 13%, respectively, when compared to the GPT3 model, as reported in [40].

### 3.3. GNSS Ionosphere and Space Weather

Ionospheric delay is an important and concerning error source in GNSS navigation and positioning systems, which has been widely analyzed [41–44]. Even though it affects the accuracy of GNSS navigation and positioning, it is well known that it can be modeled by GNSS Earth observation [45]. The ionospheric delay can be parameterized as the total

electron content (TEC) on the observation path. The relationship between ionospheric delay and TEC can be expressed as [45]:

$$I_{ion} = \frac{40.28}{f^2}\text{TEC} \qquad (4)$$

where $I_{ion}$ is the ionosohric delay and $f$ is the frequency. In the ionospheric TEC modeling, the code bias, generally represented by differential code bias (DCB), is an important error affecting the accuracy of TEC. To estimate ionospheric TEC and DCB by GNSS observations, two basic assumptions are established [45,46]. The first one is the single-layer model, in which TEC is assumed to be concentrated in a thin layer between 300–500 km. The second one is the zero-mean condition, in which the sum of satellite DCB is zero. Ionospheric modeling can be divided into three steps. Firstly, the ionospheric slant TEC (STEC) is extracted, then the STEC is transformed into vertical TEC by using an ionospheric mapping function, and finally TEC and differential code bias are estimated. Therefore, ionospheric TEC estimation and modelling are mainly focused on the following aspects, extraction of the slant TEC observables, ionospheric thin layer height and mapping function, TEC estimation and GNSS code bias handing [45,46].

The carrier-to-code leveling (CCL) method is the most used method to extract STEC [47] due to its simple implementation. A modified CCL (MCCL) method was proposed to retrieve ionospheric observables when considering the intra-day fluctuation of receiver DCB [48,49]. The MCCL method has also been extended to multi-frequency multi-GNSS [49,50]. The GF combination of GNSS observation is used in the CCL method, which affects the accuracy of TEC extraction. Therefore, the PPP method is used to improve the accuracy [51–53]. The methods of extracting ionospheric observations by PPP can be divided into single-, dual- and multi-frequency methods [54]. In addition, PPP fixed-ambiguity solutions for extracting STEC has also been proposed, instead of the more common PPP float-ambiguity solutions [55]. To take advantage of the high accuracy carrier phase observation, a method of extracting STEC by using phase observations directly was proposed [56].

Generally, for ionospheric TEC modeling, a fixed height of ionospheric thin layer and a mapping function are commonly selected [45]. However, these selections affect the accuracy of ionospheric modeling. An enhanced mapping function with ionospheric varying height was proposed in [57]. Moreover, a multi-layer mapping function was analyzed in order to reduce the ionospheric mapping errors in [58,59].

The ionospheric TEC is generally modeled through a definite mathematical function, such as spherical harmonic function, for global, generalized triangular series function, or polynomial function for regional. The DCB can be estimated together with the TEC modeling [60–63]. Moreover, the DCB can be obtained by using a global ionosphere map. Table 4 shows the mean RMS of estimated satellite DCB relative to the DCB products provided by MGEX, in which the DCB was estimated by spherical harmonic function modeling for global. Acronyms are given as: Chinese Academy of Sciences (CAS), Center for Orbit Determination in Europe (CODE), and Deutsches Zentrum für Luft- und Raumfahrt (DLR).

To handle more types of differential code bias from multi-frequency multi-GNSS, the observable-specific Signal Bias (OSB) was proposed. Several studies on the estimation and processing of GNSS OSB have been carried out [64–66].

A very interesting case study has been developed regarding plasma-spheric total electron content (PTEC). Its long-term variations have been estimated and studied from GPS observations onboard the Constellation Observing System for Meteorology, Ionosphere, and Climate (COSMIC). As it is known, the plasmasphere, or inner magnetosphere, is a region of the Earth's magnetosphere consisting of low-energy (cool) plasma and is located above the ionosphere. The long-term variation in PTEC was further analyzed using a decade-long dataset of COSMIC GPS observation data from 2007 to 2017 (Figure 5), and a high correlation was found between PTEC and the solar flux (F10.7) in the range 0.88–0.93 [67].

**Table 4.** The mean RMS of estimated satellite DCB relative to the DCB product provided by MGEX.

| System | DCB Type | CAS | DLR | CODE | System | DCB Type | CAS | DLR |
|--------|----------|-----|-----|------|--------|----------|-----|-----|
| GPS | C1C-C2W | 0.06 | 0.09 | | | C1X-C5X | 0.06 | 0.11 |
| | C1W-C2W | 0.06 | 0.16 | 0.07 | | C1X-C7X | 0.07 | 0.11 |
| | C1C-C5X | 0.07 | 0.08 | | GAL | C1X-C8X | 0.08 | 0.12 |
| | C1C-C5Q | 0.10 | 0.13 | | | C1C-C5Q | 0.10 | 0.11 |
| GLO | C1C-C2P | 0.12 | 0.12 | | | C1C-C7Q | 0.07 | 0.10 |
| | C1P-C2P | 0.14 | 0.22 | 0.15 | | C1C-C8Q | 0.08 | 0.10 |
| | C1C-C2C | 0.17 | 0.13 | | | C1X-C2X | 0.07 | 0.16 |
| BDS | C2I-C7I | 0.15 | 0.14 | | QZSS | C1X-C5X | 0.07 | 0.10 |
| | C2I-C6I | 0.19 | 0.13 | | | C1C-C2L | 0.11 | 0.12 |
| | | | | | | C1C-C5Q | 0.09 | 0.18 |

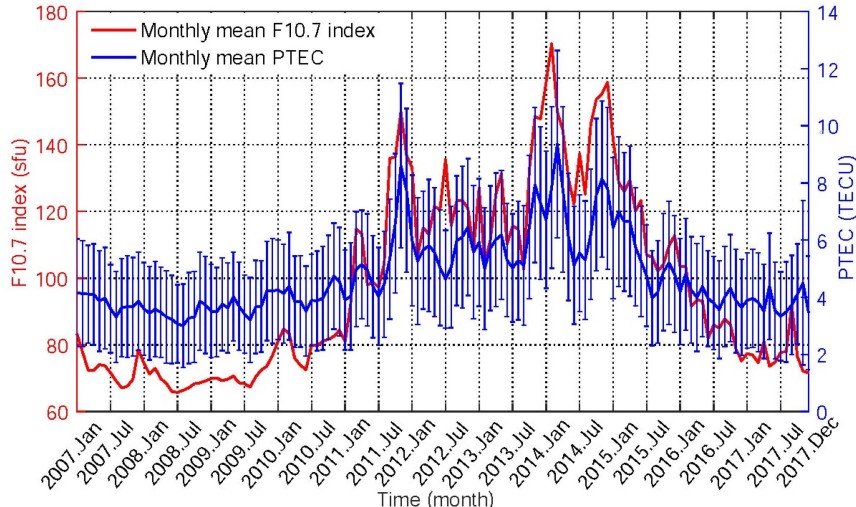

**Figure 5.** Monthly mean F10.7 index and monthly mean PTEC from January 2007 to December 2017 [67].

An analysis of the ionospheric TEC disturbances from global ionosphere maps (GIMs) was conducted during earthquakes with magnitude ≥2.5, which occurred in 2015–2018, in different latitude regions and, in particular, in A: 13° S–0.5° S (22.3° S–10° S geomagnetic), B: 0.5° S–19.5° N (10° S–10° N geomagnetic), and C: 19.5° N–32.1° N (10° N–22.5° N geomagnetic, Figure 6). The greater occurrence times of TEC decrease anomalies were observed in the southeast in Region A [68].

The thin layer ionospheric height (TLIH) was further analysed, which plays a role in mapping function (MF), and affects the accuracy of the conversion between the slant total electron content (STEC) and vertical total electron content (VTEC). In particular, a new method for determining the optimal TLIH was proposed [69], which compares the mapping function values (MFVs) from the MF at different given TLIHs with the "truth" mapping values from the UQRG global ionospheric maps (GIMs) and the differential TEC (dSTEC) method, namely, the dSTEC- and GIM-based thin layer ionospheric height (dG-TLIH) techniques. The optimal TLIH was determined using the dG-TLIH method based on GNSS data over the Antarctic and Arctic. An innovative method was recently proposed regarding multi-GNSS DCB estimation as one of the main errors in ionospheric modeling and applications. This innovative method uses independent GNSS DCB estimation (IGDE), without using the ionospheric function model and global ionosphere map (GIM). Firstly, ionospheric observations are extracted, based on the geometry-free combination of dual-frequency multi-GNSS code observations. Secondly, the VTEC of the station, represented by the weighted mean VTEC value of the ionospheric pierce points (IPPs) at each epoch, is

estimated as a parameter, together with the combined receiver and satellite DCBs (RSDCBs). Finally, the estimated RSDCBs are used as new observations, the weights of which are calculated from estimated covariances, and, thus, the satellite and receiver DCBs of multi-GNSS are estimated. Nineteen types of multi-GNSS satellite DCBs are estimated based on 200-day observations from more than 300 multi-GNSS experiment (MGEX) stations. The performance of the proposed method was evaluated by comparing it with MGEX products. The results showed that the mean RMS values were 0.12, 0.23, 0.21, 0.13, and 0.11 ns for GPS, GLONASS, BDS, Galileo, and QZSS DCBs, respectively, with respect to MGEX products, and the stabilities of estimated GPS, GLONASS, BDS, Galileo, and QZSS DCBs were 0.07, 0.06, 0.13, 0.11, and 0.11 ns, respectively, as reported on in [70].

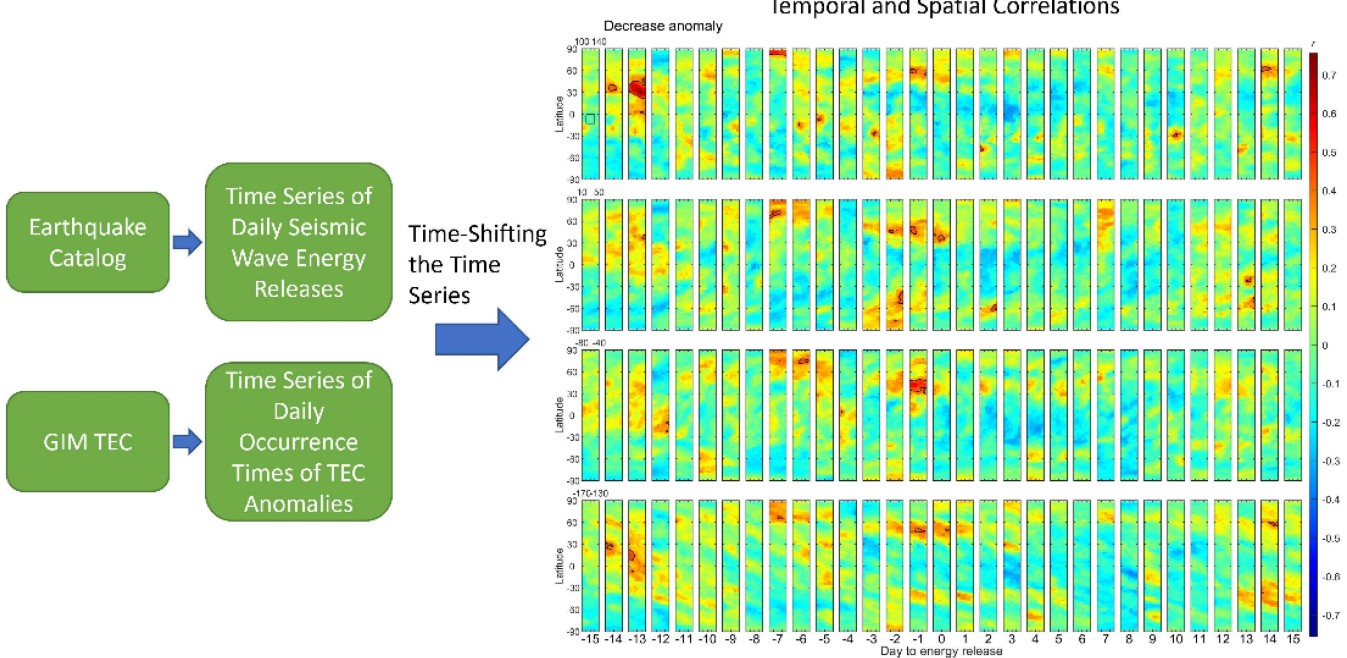

**Figure 6.** Temporal and spatial correlations [68].

Other interesting progress concerns the greater opportunities for positioning offered by Beidou (BDS) GNSS with precise ionospheric delay corrections. The BDS reflection signal detects multiple moving targets under multiple-input and multiple-output (MIMO) radar systems and proposes a series of methods to suppress multiple Doppler phase influences and improve the range detection property. The simulation results showed restored target peaks, which matched the RCS data more accurately, with the GNSS-R Doppler phase influence removed, which proved the proposed method could improve target recognition and detection resolution performance [71]. Many Differential Code Biases, DCBs and DCB types of the new BDS-3 signals from 30-days Multi-GNSS Experiment (MGEX) observations, were estimated and investigated. Compared with the DCB values provided by the Chinese Academy of Science (CAS) products, the mean bias and root mean squares (RMS) error of the new BDS-3 satellite DCBs were within ±0.20 and 0.30 ns, respectively. The satellite DCBs were mostly within ±0.40 ns with respects to the Deutsches Zentrum für Luft- und Raumfahrt (DLR). The four sets of constructed closure errors and their mean values were within ±0.30 ns and ±0.15 ns, respectively. The mean standard deviation (STD) of the estimated satellite DCBs was less than 0.10 ns (Figure 7). Of particular note was the fact that the estimated satellite DCBs were more stable than DCB products provided by CAS and DLR [72].

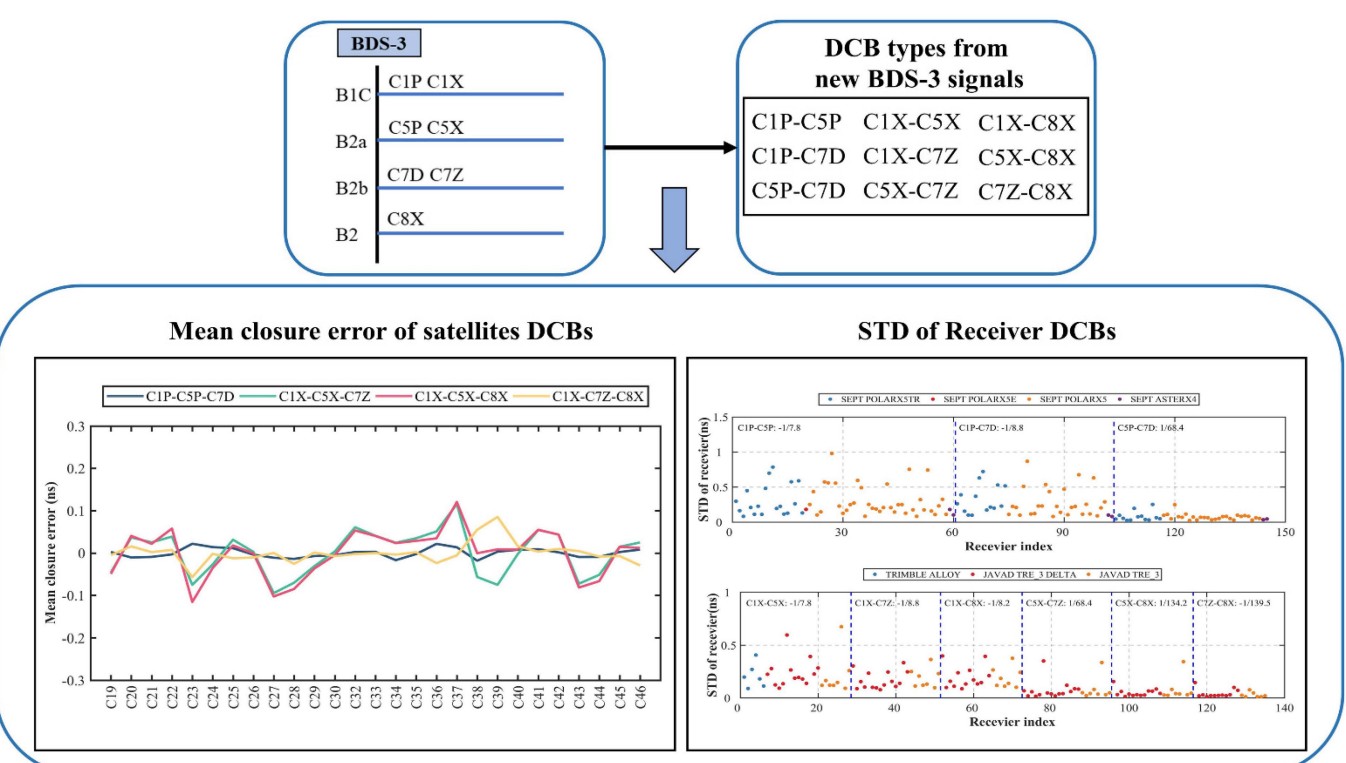

**Figure 7.** Mean closure error of BDS-3 satellites DCBs and STD of recilver DCBs [72].

### 3.4. GNSS-Reflectometry

GNSS reflectometry (GNSS-R) is a very useful tool for remote sensing and plays a key role in various applications. Signals reflected from the Earth's surface are analyzed to measure various geophysical parameters. Multipath delay is one of the main error sources in GNSS navigation and positioning. The changes in the polarization characteristics, amplitude, phase, and frequency of the reflected signal reflect the roughness of the reflecting surface, such as changes in coastal water levels and snow thickness. Therefore, accurate estimations of the multipath delay can invert the physical properties and geophysical parameters of the reflecting surface. One of the available solutions is to use the upward-facing GNSS right-handed antenna to receive the direct signal and the downward-facing GNSS left-handed antenna to receive the reflected signal. Using GNSS precise single-point positioning, the delay of the direct and reflected signals can be estimated, and then the water level and snow thickness changes can be inversed. Another solution is to use the signal-to-noise ratio (SNR) of GNSS observations to estimate soil and snow thickness changes, but not all receivers have SNR observations. Qian and Jin [73] used geometry-free linear combination of GNSS code and carry phase observations (L4 observations) to estimate snow thickness. The L4 observations were not affected by geometrical factors and contained multipath residuals, which could effectively represent the multipath. The snow thickness was inverted according to the relationship between the changes of the L4 observations and the measured snow thickness. The snow thickness obtained by using the L4 observations was in good agreement with the results obtained from the signal-to-noise ratio data. However, a single system sometimes has limited observation satellites, and multiple systems improve space coverage. Qian and Jin [73] combined GPS and GLONASS observations from the IGS station GANP (Slovenia) to estimate the snow thickness variation in 2012 and 2013 from GNSS geometry-free linear combination observations (L4 observations) and signal-to-noise ratio (SNR), respectively (Figure 8). The accuracy was improved for a single system, but the L4 method needed to be further improved when compared to the SNR method, because of ionospheric delay and other unremoved errors.

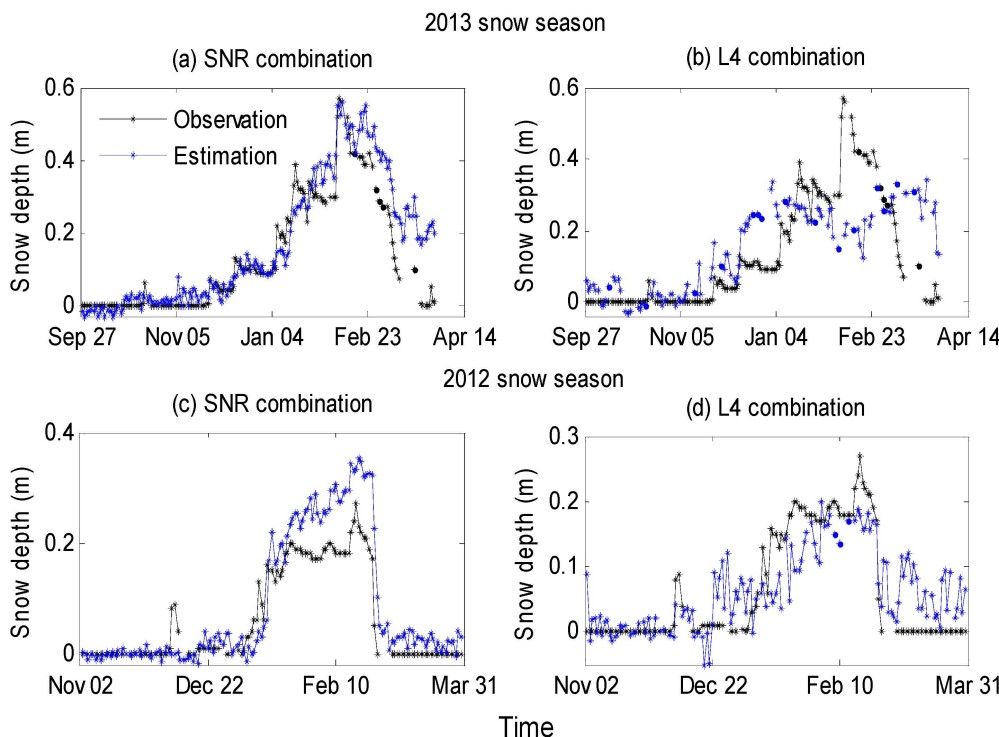

**Figure 8.** Snow thickness changes at IGS station GANP in 2012 and 2013 from the combined GPS and GLONASS observations with geometric-free combination (L4) and signal-to-noise ratio (SNR), respectively [73].

### 3.5. Earthquake Monitoring by GNSS

Traditional earthquake magnitude and rupture are inverted using seismograph or accelerometer observation data. However, seismograph data often has missing saturation and data, which cannot completely record the seismic co-seismic displacement amplitude. Although the accelerometer data is not missing, the seismic displacement obtained by integrating the accelerometer data can be distorted by the tilt and rotation of the instrument. Nowadays, GNSS can perform precise single-point positioning and estimate absolute seismic displacement, as well as inverse the rupture, with high accuracy. For example, using the 1 Hz GPS, BDS, GLONASS and Galileo observation data at the LASA station provided by Beidou Experiment Tracking Station (BETS), the displacement of the 25 April 2015 Mw 7.8 Nepal earthquake was estimated and compared with the strong-seismic records near the Tibet area. Figure 9 shows acceleration, velocity, and displacement time series based on constant velocity (CV) dynamic PPP (CVDPPP) with BDS single system, GPS single system, BDS + GPS dual system, GPS + GLONASS dual system and BDS + GPS + GLONASS + Galileo four system during the 300 s period of the initial earthquake occurrence [74]. GNSS results showed high consistency with the displacement time series obtained by the strong motion instrument. The seismic waves estimated by CVDPPP were not affected by the distortion caused by the rotation and tilt of the instrument. By combining multi-system observation data, the velocity and acceleration waves obtained by the CVDPPP model were smoother, which also verified the advantages of multi-system GNSS in monitoring earthquake co-seismic displacements.

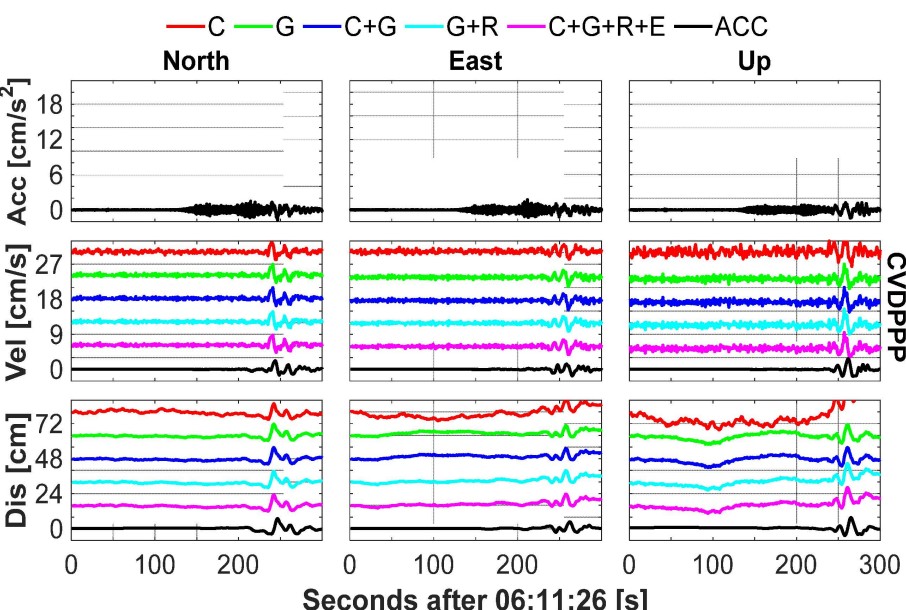

**Figure 9.** Acceleration, velocity, and displacement time series based on constant velocity (CV) dynamic PPP (CVDPPP) for BDS, GPS, BDS + GPS, GPS + GLONASS and BDS + GPS + GLONASS + Galileo four system during 300 s after UTC 06:11:26 [74].

### 3.6. GNSS Integrated Techniques for Land and Structural Health Monitoring (SHM)

Real-time monitoring of the engineering structures safety is necessary, since these structures play economic, social and environmental roles. The assessment of the coherence between the expected displacements and those affecting the structures could provide feedback about their behavior in the elastic field, complying with safety regulations. GNSS monitoring systems, used in combination with geotechnical, hydraulic, and structural systems, could allow the monitoring of real-time displacements, with high accuracy, even remotely.

Among the emerging applications of GNSS, those related to monitoring through low-cost mobile smartphone type instruments play a very important role because of the wide availability of these instruments. With Xiaomi Mi8, the first GNSS dual-frequency smartphone embedded with the Broadcom BCM47755 GNSS chip, tests using both VADASE (Variometric Approach for Displacement Analysis Stand-alone Engine) and VARION (Variometric Approach for Real-Time Ionosphere Observations) algorithms were even able to derive real-time STEC variations [75]. Other authors using smartphone accelerometer (Bosch BMI160) and a low-cost dual frequency GNSS reference-rover pair (u-blox ZED-F9P) have achieved high precision values ($\sigma$) of $\pm 7.7$, 8.1 and 9.6 mm in the East, North and Up (ENU), respectively, which were comparable with the declared precision potential ($\sigma$) of the smartphone accelerometer of $\pm 8.8$ mm [76].

The interaction between GNSS and Remote Sensing produces excellent results for the monitoring of strategic structures, such as dams. Dam structures can be monitored via traditional contact sensors (extensometers, accelerometers, tiltmeters), ground-based methods (ground-based SAR, ground-based photogrammetry, terrestrial laser scanning, robotic total stations), and GNSS. Remotely based methods include airborne Light Detection and Ranging (LiDAR) and space-borne InSAR. The pros and cons of these methods are summarized in [77].

Furthermore, other new techniques have been developed and tested. The first of these was based on an unsupervised classification and was suitable for automating of the process. The second was based on visual matching with contour lines, with the aim of fully exploiting the dataset. Their performances were evaluated by comparison with water levels measured in situ ($r^2 = 0.97$ using the unsupervised classification, and $r^2 = 0.95$ using visual matching), demonstrating that both techniques were suitable to quantify reservoir surface extension. However, ~90% of available images were analyzed using the visual

matching method, and just 37 images are out of 58 using the other method. The evaluation of the water level from the water surface, using both techniques, could be easily extended to un-gauged reservoirs to manage the variations of the levels during normal operation. In addition, during the period of investigation, the use of GNSS allowed the estimation of dam displacements. By comparing results from both techniques, relationships between the orthogonal displacement component via GNSS, estimated water levels via remote sensing and in situ measurements were investigated. In fact, the moving average of the displacement time series (middle section on the dam crest) showed a range of variability of $\pm 2$ mm (Figures 10 and 11). The dam deformation versus reservoir water level behavior was different during the reservoir emptying and filling periods, indicating a kind of hysteresis loop [78,79].

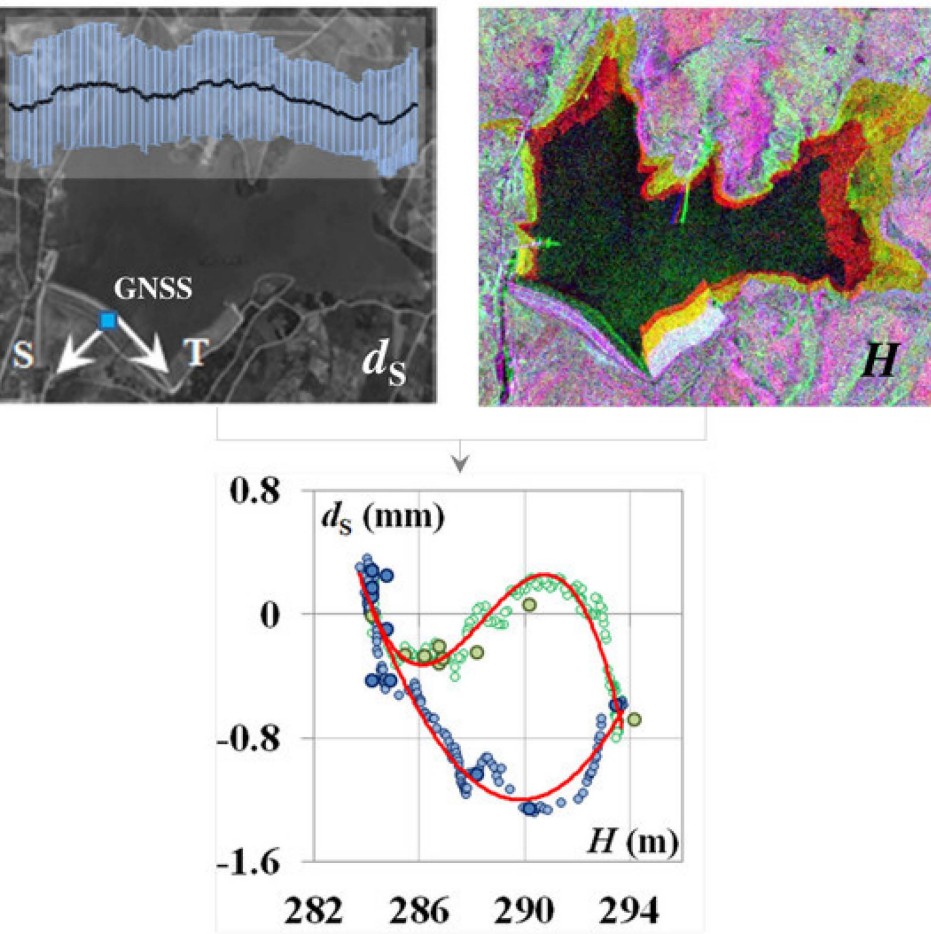

**Figure 10.** Water surface and level of a reservoir from different remote sensing approaches and comparison with dam displacements evaluated via GNSS [78]. Top left: The Castello dam on Magazzolo reservoir (37°34′51″ N, 13°24′48″ E, WGS84) with local reference system (T and S indicate directions tangential and orthogonal to the dam) and temporal behavior of timely averaged GNSS displacements of the central section along S-axis (black dots) with over imposed $\pm$ the standard deviation within the 2-month moving window (blue bars). The black dot indicates $d$s occurring with minimum the water. Top right: diachronic CSK false-color composition for increasing Hm, (281.10, 285.31 and 291.40 m represented in red, gran and blue color scales, respectively). Downwards: He, and Hm, vs. $d$s. during the emptying (green circles) and filling (blue circles) phase, over-imposed two interpolation curves (red continuous lines).

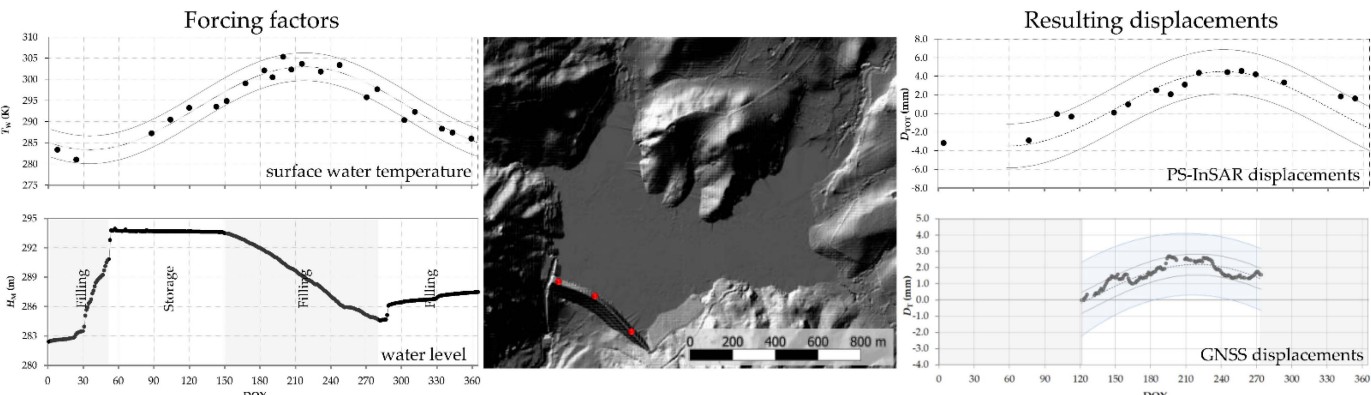

**Figure 11.** Comprehensive dam monitoring with on-site and remote-retrieved forcing factors and resulting displacements [79]. Top left: Daily water levels measured in situ, $H_{IS}$ (m a.s.l.). A linear interpolation curve is superimposed–imposed (continuous line); Up left: the temporal behavior of the water surface temperature, $T_W$ (K), retrieved by Landsat 8 data, using both 189-034 and 190-034 scenes. A sinusoidal interpolation curve (continuous dashed line) and a confidence band of $\pm 2 \times \sigma_{XY}$ between measured and interpolated values (dark grey lines) are reported to facilitate the interpretation of the phenomenon. Center: hill shade of the dam–reservoir system derived from a digital surface model by setting azimuth and elevation of the incident light at 315 and 45 degrees, respectively). Superimposed the positions of the GNSS stations (red points). Top right: the temporal behavior (DOY in x-axis) of the horizontal total displacements orthogonal to the dam ($D_{TOT}$, mm) is estimated via PS–InSAR (red dots). A sinusoidal interpolation curve (continuous dashed line) and a confidence band of $\pm 2 \times \sigma_{XY}$ between measured and interpolated values (dark grey). Up right: the temporal behavior (DOY in x-axis) of the horizontal total displacements orthogonal to the dam ($D_{TOT}$, mm) is estimated via Global Navigation Satellite System (GNSS) (grey dots). A sinusoidal interpolation curve and confidence band of $\pm 2 \times \sigma_{XY}$ between measured and interpolated values (dark grey lines) were reported to facilitate the interpretation of the phenomenon (continuous line). The interpolation curves at the 10th and 90th percentiles of the raw data were reported (pale blue band) as a measure of its variability.

Many studies have focused on the SHM of other composite structures, such as bridges. Yigit and Gurlek [80] proposed some testing of a high-rate GNSS precise point positioning (PPP) method for detecting dynamic vertical displacement response. In particular, the usability of PPP for evaluating the dynamic displacement response of a structure was analyzed with different experiments on cantilever beam structures. Four cases with different vibration frequencies between 0.94 and 2.90 Hz were selected to compare the PPP and relative precise methods in the time, position and frequency domains. In addition, the effects of the ultra-fast products and the final precise orbits on the PPP kinematic solution, in terms of vertical oscillation detection, were examined. Xi et al. [81] used the BDS/GNSS system to conduct an experiment on a bridge in China to evaluate the performance of BDS through comparing with GPS, and found that the accuracy of BDS in static mode could be up to 2–3 mm and 5–7 mm in the horizontal and vertical components. With the monitoring data of a bridge, BDS had the same, or even better, monitoring performance and data quality as GPS. Meng et al. [82] carried out a system for large bridge monitoring, while Xi et al. [81] proposed a multi-GNSS integration processing method and presented a case study on bridge monitoring using multi-GNSS observations (BDS, GPS and GLONASS) with high cutoff heights. Based on the experiments conducted, it was shown that, with more available satellites and stronger satellite geometry, the GPS/BDS/GLONASS combination showed the highest accuracy with 1–2 mm horizontal accuracy and 2–5 mm vertical accuracy. With the integration of GPS/BDS/GLONASS, different cutoff values were set in the data processing in the bridge monitoring application. The results showed that the accuracy in the horizontal component could always reach 1–2 mm with increasing cutoff elevation angles (10° to 40°), even when the upper limit of 40° was selected [83]. More

recently, Manzini et al. [84] evaluated the use of low-cost GNSS stations for SHM, through different combinations of GNSS receivers and antennas. Several sets of parameters and processing requirements were also evaluated using the open-source software RTKLib. The performance of the proposed solution was evaluated through two dynamic experimental scenarios, and results showed its ability to track rapid displacements of up to 4 mm and oscillations of 1 cm with a frequency of up to 0.25 Hz with a 1 Hz receiver. Finally, a two-week dataset, acquired from a network of low-cost GNSS stations distributed on a suspension bridge, was used to validate the in-situ performance. The results showed good agreement between GNSS time series, conventional displacement sensors and numerical simulations [84]. A very interesting review has been published in recent years concerning the use of GNSS with Based Dynamic Monitoring Technologies for SHM. For more details the reader should refer to the work [85].

*3.7. GNSS Congruence with Different Modes' Solutions (NRTK, PPP, Static)*

Other GNSS emerging applications are related to uses with different modes' solutions, such as Network Real Time Kinematic (NRTK) or PPP, or also static. As an example, Baybura et al. [86] suggested examining the accuracy of Network RTK (NRTK) and Long Base RTK (LBRTK) methods with repetitive measurements. The NRTK and LBRTK measurements were performed on different days between 2015 and 2018 with various survey campaigns, and considered the results of the static measurements as true coordinates. The results of the NRTK and LBRTK methods showed that the LBRTK and NRTK methods provided similar results for baseline lengths up to 40 km, with differences of less than 3 cm horizontally and 4 cm vertically. Lu et al. [87] compared static and dynamic PPP measurements, and the numerical results during the static and dynamic tests showed that the proposed positioning study could achieve a positioning accuracy of a few centimeters within one hour. As expected, the positioning accuracy was significantly improved by the combination of GPS, BeiDou and Galileo as a result of the larger number of satellites used and the more uniform geometric distribution (DOP) of the satellites.

The evaluation of the coordinates from a GNSS survey (in Network Real Time Kinematic, Precise Point Positioning, or static mode) has been analyzed in several scientific and technical applications, and many have been carried out to compare Precise Point Positioning (PPP), Network Real Time Kinematic (NRTK), and static modes' solutions, using the latter as the true, or the most plausible solution. Another study has been developed using the Italian GNSS CORS of Sicily (Figure 12) to compare the GNSS survey methods mentioned above, using some benchmark points. The tests were carried out by comparing the survey methods in pairs to check their solution congruence. The NRTK and the static solutions were referred to a local GNSS CORS network's analysis. The NRTK positioning was obtained with different methods, such as VRS, Flächen-Korrektur-Parameter (FKP), Nearest (NEA) and the PPP solution, and calculated with two different software (RTKLIB and Canadian Spatial Reference System CSRS-PPP). A statistical approach was performed to check if the distribution frequencies of the coordinate's residual belonged to the normal distribution for all pairs. The results showed that the hypothesis of a normal distribution was confirmed in most of the pairs and, specifically, the static vs. NRTK pair seemed to achieve the best congruence. Involving the PPP approach, the pairs obtained with CSRS software achieved better congruence than those involving RTKLIB software [88].

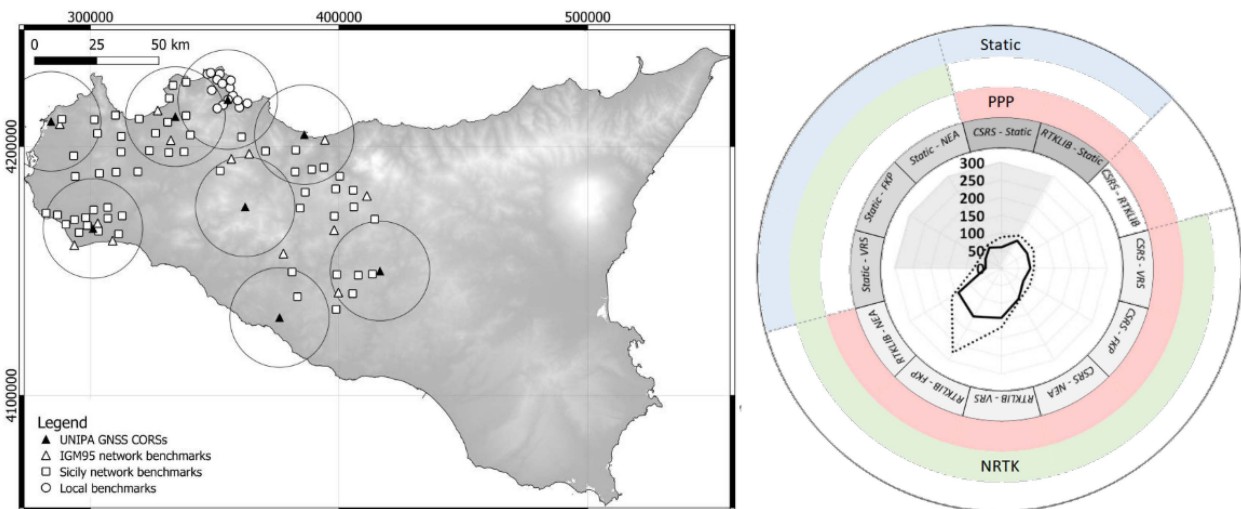

**Figure 12.** Different solutions for GNSS NRTK, PPP or static positioning. Left: UNIPA GNSS CORSs (black triangles) and GNSS reference benchmarks (IGM95 network benchmarks with white triangles; Sicily network benchmarks with white squares; and local benchmarks with white circles). 20 km buffer circles from the GNSS CORS are shown. Reference system UTM-WGS84 33N (ETRF2000-RDN2008)-EPSG6708. Right: Δh differences, pre- and post-outliers' removal (dashed and continuous lines, respectively) [88].

## 4. Discussion and Conclusions

A detailed review on multi-GNSS for Earth Observation and the progress of its emerging applications has been presented in this paper. One of the main findings from this study is that, nowadays, the GNSS technique is involved not only in traditional positioning applications, but more widely for remote sensing applications, which represents one of the most used techniques in the field of Earth observations. As it has been deeply discussed in the paper, with continuous improvements and developments, in terms of performance, availability, modernization and hybridizing, multi-GNSS could become a milestone in the future applications. So, what scenarios could we be looking at in the coming years? The answer to this question is quite difficult, but, obviously, there are various fields of multi-GNSS applications to be explored and analyzed. One of these is the use of *GNSS for Autonomous Space Navigation*, since the use of GNSS for this purpose is crucial for space missions. In fact, it can be performed directly on-board and in real time to enable autonomous guidance with reducing or avoiding the delays in Earth-to-space communications and lack of signal coverage. For this purpose, different technologies (GNSS and IMU) and algorithms can be used with enabling, e.g., the precise spacecraft formation flights and landings required in these operational areas.

Another investigation concerns *GNSS atmospheric modeling*. It is widely known that multi-GNSS are influenced by the Earth's atmosphere, including the ionosphere (electrically charged), and the troposphere (neutral atmosphere), modeled with complex mathematical equations [89–91]. Although atmospheric effects on GNSS signals are annoying parameters for positioning and navigation applications, they can provide valuable information for many applications, such as natural hazard monitoring or weather forecasting. Therefore, the modeling of atmospheric effects on multi-GNSS positioning applications (GPS, GLONASS, Galileo, BDS, QZSS and IRNSS) on the ground and in space needs to be deeply analyzed so as to improve ZTD and TEC modeling at regional and global scales, as well as scintillation and forecast models, ionosphere models and tropospheric gradient models.

Further interesting applications of satellite navigation are focused on *GNSS-R* emerging applications, such as methods and measurements techniques for *Remote Sensing of Soil Moisture Content (SMC)*. As it is known, SMC plays an important environmental role in the assessment of climate change and environmental monitoring of areas prone to flooding,

drought and evapotranspiration. SMC also allows the monitoring of water runoff and surface erosion. By correlating other environmental variables, such as land surface temperatures, land cover or precipitations, SMC is commonly used as an input parameter for many climate models. In agriculture, SMC is a crucial indicator of plant growth and crop yield. In recent decades, satellites closer to Earth equipped with GNSS-R receiver, active microwave (ALOS-2, Sentinel-1, TerraSar-X) or passive microwave (AMSR2 and SMOS), have provided an opportunity to detect SMC from space using a wide range of techniques and sensors. A scientific treatment of the issues related to SMC (e.g., GNSS-R Techniques, Methods, and Applications) has already recently developed with wide potential applications [92,93]. Another interesting investigation could be *GNSS-R Earth Remote Sensing from SmallSats* and substantial economic development investments were carried out in so-called small satellites in recent years, such as BuFeng-1 [94], CYGNSS [95], Fengyun-3 series [96], FSS-Cat [97], HydroGNSS [98], PRETTY [99], and Spire CubeSats series [100]. Small satellites are changing the Earth remote sensing parameters by exploiting innovative payloads. Thus, the spatiotemporal sampling properties of GNSS-R could create new scenarios for studying, specifically devoted to wind speed determination, SMC determination, vegetation water content monitoring, and supporting sustainable soils developments. Additionally, GNSS-R can be used for *ocean monitoring*, as presented in the work of [101–104].

In addition, GNSS can provide meaningful support to *Precision Farming* (PF). PF has been widely implemented in almost all agricultural production systems over the past 20 years. Obviously, PF developments differ in the world according to technological, agronomic, economic and cultural differences existing between countries. PF has been widely used in developed agricultural countries. Considering benefits and limits of increasing PF adoption all around the world, GNSS could provide innovative methods and applications to optimize operating modes, particularly in developing agricultural economies [105–108].

The use of *GNSS* for *forest and wetland hydrology* is also developing. According to other geomatics techniques (Remote Sensing, UAV/UAS and LiDAR), it can be considered a resource for the assessing of climate-related environmental risks, such as fires, landslides, epidemics of forest diseases, rapid deterioration of the quality of watercourses, and conversion of forest wetlands to montane forests, due to the deposition of eroded soil [109–111]. Newer ECOSTRESS [112] and SMAP [113] satellite types, specifically designed to obtain soil moisture information with dense forest cover, may be an important improvement in this emerging study.

The use of ground-based GNSS and/or radio occultation techniques can also be useful in the field of *natural hazards*, such as those related to the emission of hazardous gases and ash into the atmosphere from *volcanic clouds*. Analysis of Signal Noise Ratio (SNR) data can demonstrate daily repeatability and seasonal trends indicating the strong dependence of multipath error on changes in the antenna environment, but can also be an indicator of sudden changes in volcanic cloud composition and height [114–117]. *GNSS interference* detection and spoofing provide an important area of exploration, since multipath and Non-Line-Of-Signals (NLOS) are the main errors occurring in different GNSS applications, e.g., civil (urban transport applications) and military uses [118–122].

Last, but not least, the massive deployment of multi-frequency *GNSS CORS* at global, regional, and national scales has allowed the continuous use of time series data that is mostly free of charge. Moreover, regarding the upcoming CORS upgrades to quadric stellation (BDS, Galileo, GLONASS, GPS), these infrastructures can provide unlimited potential for both technical and scientific applications, e.g., the evaluation of a global reference system and its inconstancy, geodynamic analysis, PF, mining, SHM, surveying and land cadastral management, soil moisture mapping, drought, snow depth, airborne UAV, road and rail transport and logistics, maritime navigation and aviation [123–126].

**Author Contributions:** Conceptualization of the Manuscript Idea: S.J. and G.D.; Methodology and Software: Q.W. and S.J.; Supervision and Funding Acquisition: S.J. and G.D.; G.D., Q.W. and S.J. wrote the original draft preparation; S.J. and G.D. reviewed and edited this paper. All authors have read and agreed to the published version of the manuscript.

**Funding:** This research was funded by the National Natural Science Foundation of China (NSFC) Project (Grant No. 12073012), Shanghai Leading Talent Project (Grant No. E056061) and Jiangsu Natural Resources Development Special Project (Grant No. JSZRHYKJ202002).

**Data Availability Statement:** The BDS observation data from IGS MGEX networks can be obtained https://cddis.nasa.gov/archive/gps/data/daily, last accessed on 31 July 2022.

**Acknowledgments:** The authors thank IGS for providing the BDS observation data of MGEX networks. The authors would like to express their sincere thanks to Claudia Pipitone for her valuable work in revising the English throughout the text.

**Conflicts of Interest:** The authors declare no conflict of interest.

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
