# Peer review of "A Review on Multi-GNSS for Earth Observation and Emerging Applications"

_remotesensing, doi:10.3390/rs14163930_

Round 1

Reviewer 1 Report

The goal for the paper is stated as “The aim of this review is to present the latest state of the multi-GNSS art findings,”   As I stated in my previous review the goal is very broad with selected examples shown without justifying the selections.   There is still no justification.     Nowhere can I find in the paper the most simple justification of studying “multi” GNSS concepts or applications.  Why multi versus single constellation?   It might seem obvious, such as the greater number of satellites above the horizon for a position fix…. But the authors do not tell us this.   They never mention the constellations use slightly different geoidal models. 

What is missing from paper is information on the non-civil signals.  For example, GPS has signals of better quality that are available to the military use in both the USA and NATO, but also to non-military governmental users.  I know this is true for the other GNSS constellations.  But there is no mention in the text.  The authors should acknowledge the other uses and state their manuscript will only discuss publically available signals.

Recheck the number of satellites “in orbit” for each constellation.  For example, there are 32 GPS satellites in orbit but only 31 satellites are operational as of 3 August 2022.

“Nowadays, the multi-frequency multi-GNSS technology is used for 84 earth observation.”  This statement is not necessarily true.  The FCC (USA agency) has strict requirements for using only GPS for 911 calls unless the producer of the device (e.g. Samsung, Huwaii, etc.) can demonstrate the reliability of using other constellations.   I suspect other agencies in other countries have similar regulations.

Section 2.3 GNSS Positioning Methods

The authors begin this section by noting 2 methods, both of which are theoretically based on differential positioning: “There are two common methods of GNSS positioning, namely differential position-ing, and precise single point positioning (PPP).”  They never mention code-based positioning, the most widely used positioning method.  And yes, code-based solutions can be based on multi-GNSS constellations.

Grammar:

I noted this before and will not elaborate again but the manuscript is not written well.  Just a few examples:

“has been fastly developed“

 “and attracted 34 more attentions”

 “however, p the base stations”

 “has reached more attentions,”

 “Nowadays multi-GNSS represents one of the best technique in the field of Earth observations. “    Remote sensing from drone, airborne, satellite is considerably more well-known and utilized. 

 “These stations provide sufficient guarantee”  -- what does this mean?

 Table 3. Information of GNSS receiver in MGEX”

 Infact Bevis”

 “Remotely based methods include airborne Laser Imaging Detection and Ranging (LiDAR)”  LiDAR is defined as Light Detection and Ranging.

 “that, nowadays, GNSS technique,”

 “infact,”

 “is also growing up,”

2.5 Multi-GNSS Observations

This section discusses one association that collects/disseminates GNSS observations.   There are others that are also public (e.g. JPL, COOPS) but the most widely used for commercial applications are from private vendors (e.g. Trimble).  The authors should include this as the commercial solutions will continue to provide the most accurate and highest quality data in real-time.

Noted in first review and not addressed in revision:

Also, be consistent in defining the acronyms.   You define GNSS, BDS and PNT but not GLONASS and GPS.

Figure 3.  The graphs clearly show the errors are large at UTC 0 time but decrease with increasing hour.   The text does not note this or provide an explanation.   I believe these data demonstrate the errors decrease with increasing observation time.

Author Response

Please see file in attach.

Best regards

Gino Dardanelli

(corresponding author)

Reviewer 2 Report

Dear Authors,

Thank you for effectively addressing the suggested comments in the revised version of the manuscript.

With best regards

Author Response

(The authors gave the same response as above.)
